# The Role of Family Health in Mediating the Association between Smartphone Use and Health Risk Behaviors among Chinese Adolescent Students: A National Cross-Sectional Study

**DOI:** 10.3390/ijerph192013378

**Published:** 2022-10-17

**Authors:** Fangmin Gong, Zhaowen Lei, Zhuliu Gong, Hewei Min, Pu Ge, Yi Guo, Wai-Kit Ming, Xinying Sun, Yibo Wu

**Affiliations:** 1School of Literature and Journalism Communication, Jishou University, Jishou 416000, China; 2School of Public Health, Peking University, Beijing 100871, China; 3Bachelor of Pharmacy Institute of Chinese Medicinal Sciences, University of Macau, Macao 999078, China; 4Department of Infectious Diseases and Public Health, Jockey Club College of Veterinary Medicine and Life Sciences, City University of Hong Kong, Hong Kong 999077, China

**Keywords:** family health, smartphone use, adolescents, health risk behaviors, mediating role, structural equations

## Abstract

The direct impact of smartphones on health risk behaviors of adolescent students has been verified. However, the mediating mechanisms that underly this relationship remain largely unknown. Therefore, the aim of the study is to explore the role of family health in mediating the relationship between the frequency of smartphone use and adolescent students’ health risk behaviors. A questionnaire was used to collect cross-sectional data from 693 adolescent students aged 12–18 in China and a structural equation model was analyzed. Among the nine health risk behaviors, the most frequent health risk behaviors in Chinese adolescent students were non-compliance walking behaviors (M=Mean; SD = Standard deviation) (M ± SD) (2.78 ± 1.747), eating unhygienic food (M ± SD) (2.23 ± 1.299), being subjected to physical violence (M ± SD) (2.19 ± 0.645)*,* and leaving home (M ± SD) (2.13 ± 0.557). The SEM results showed that the adolescent students’ smartphone use had a positive impact on delaying the age of first alcohol consumption (*β* = 0.167, CI:0.067 0.287) and a negative impact on the non-compliance walking behaviors (*β* = 0.176, CI:0.011 0.266). Family health plays an indirect-only mediated role (the proportions of indirect-only mediated roles are 11.2%, 12.4%, and 11.5%) in the relationship between smartphone use and adolescent students’ partial health risk behaviors: (CI: −0.042 −0.002), (CI: −0.049 −0.005), and (CI: −0.043 −0.002). These findings provided a theoretical and practical basis for better interventions in adolescent health risk behaviors.

## 1. Introduction

More and more problems related to excessive smartphone use have emerged worldwide, especially among adolescents, who constitute the largest group of users [1,2,3]. In the past decade, the reports that smartphones cause adolescent health risk behavior make researchers put more effort into the relationship between the smartphone use and the health risk behaviors. The health risk behaviors in adolescents refer to the behaviors that cause direct and indirect harms on health and well-being during their adolescence and adulthood [4]. According to the classification of Centers for Disease and Prevention and Control’s Youth Risk Behavior Surveillance System (YRBSS), teenagers’ health risk behaviors include the following six main categories: behaviors that lead to unintentional injuries and violence, smoking, alcohol, and other drug use, risky sexual behaviors, unhealthy eating habits, and lack of exercise [5]. The effects of smartphone use on dangerous behavior may be particularly strong in the adolescent community [6]. According to the Facebook influence model [7], social media environments allow adolescents to express their sense of identity or an idealized self to the audience, and encourages them to display abnormal behaviors, such as risk-taking behaviors, on Facebook. The media practice model [8] further hypothesizes that the use of social media reflects adolescents’ current sense of identity and the person they desire to be [9], and it is often expressed as engagement in and the display of risk behaviors on social media platforms [10].

However, some scholars believe the impacts of smartphones on adolescent health are complex. There are both positive and negative effects [11,12]. Many researchers have previously connected the smartphone use to important aspects of the central social development of adolescent, such as self-esteem, social bonding, peer victimization, internalization problems, criminal behavior, and sexual self-exploration [13,14,15,16,17,18]. The existing evidence shows that smartphones not only contribute to the daily use of adolescents, but also facilitate screening, treatment, and prevention of disease [19,20,21,22]. It also simultaneously produces harmful influence. For example, risk factors affecting the physical health of adolescents, such as sleep deprivation of adolescents, drowsiness, and obesity, are related to the over-use of smartphones [23,24,25].

Some scholars have found that smartphones are not only used to promote family communication and connection [26], but also distract high-quality time spent together and lead to family conflict [27]. Not all parents agree that adolescents’ digital technology involvement is a net negative for family relationships. In a national study of 1240 U.S. parents, 18 percent of reports show that children’s mobile devices benefited family relationships. In contrast, 15 percent of reports show that smartphones were harmful to family relationships, and 67 percent of reports show that the difference was weak [28].

The risk behaviors usually emerge during the adolescent period, which cover the behaviors that make adolescents get injured or worse, such as drug use, unsafe sexual behaviors, and violence-related behaviors (Centers for Disease Control and Prevention) [5,29]. According to the World Health Organization (WHO) survey data on adolescent health risk factors, the prevalence of alcohol consumption among Chinese adolescents aged 15–19 is as high as 41.20%. The proportion of adolescents aged 13–17 who have experienced violence in the past 30 days or more is on average 26% in low-income countries [30]. In most countries, two-thirds of children are subjected to violent discipline by their guardians [31]. The increase of adolescents’ health risk behaviors and the negative impacts receive extensive attention from society; thus, it is necessary to discuss the triggering factors and the underlying mechanism, which may contribute to intervention and prevention programs for adolescents’ health risk behaviors.

The evidence suggests that the impact of smartphones on adolescent health risk behaviors may vary depending on the role of other factors, such as family health. In fact, the behaviors and family environment created by parents have a direct impact on the emergence and transformation of adolescents’ behavior, regardless of whether those behaviors lead to unhealthy eating habits, early sexuality, online game addiction, substance addiction, or numerous other health risk behaviors [32,33,34,35]. In addition, early family intervention is an effective source of influence in preventing the onset of health risk behaviors in adolescents and is particularly effective at the level of substance addiction. A study suggests that family interventions may reduce smoking rates from 16 to 32 percent among adolescents when they are 11 to 14 years old [36]. Family health resources are also key factors that influence the occurrence of adolescent health risk behaviors, for example, adolescents in low-income families are more likely to be affected by unhealthy diets [37].

Family health may moderate the effects of media exposure on adolescent risk behaviors [38]. Some studies have shown that active parental involvement can be effective in preventing the potential dangers of digital media for adolescents [39]. Pediatricians say that when parents and children watch videos together and make this behavior become a way to discuss important family values, the positive impact on adolescent health will be evident [40]. However, the mediating role of family health has often been overlooked in media effects research.

### Study Purpose and Hypotheses

This study focuses on building a structural equation model to explore the role of family health in mediating the relationship between the frequency of smartphone use and adolescent students’ health risk behaviors, and to explore the role of smartphone use as a direct influence on adolescent students’ health risk behaviors (the hypothesized model was shown in Figure 1).

In summary, our hypotheses are as follows:

**H1.** 
*The frequency of smartphone use predicts health risk behaviors significantly and positively.*


**H2.** 
*The frequency of smartphone use will have a significant positive predictive effect on family health.*


**H3.** 
*Family health would be a significant negative predictor of the occurrence of health risk behaviors in adolescent students.*


**H4.** 
*Family health mediates the relationship between the frequency of smartphone use and health risk behaviors.*


## 2. Materials and Methods

### 2.1. Participants and Procedures

We adopted a multi-stage sampling method from 10 July 2021 to 15 September 2021. There were 23 provinces, the provincial capitals of 5 autonomous regions, and 4 municipalities directly under the central government (Beijing, Tianjin, Shanghai, and Chongqing), for a total of 120 cities. Based on the data of “Result of the Seventh National Census in 2021” [41], 120 urban residents were sampled by quota (the quota attributes are gender, age, urban–rural distribution), and the samples obtained for gender, age, and urban–rural distribution matched the demographic characteristics, basically. Finally, 11031 questionnaires were valid, with an effective rate of 94.2% as shown in Figure 2. This study scheme was approved by the Institutional Review Committee of Ji’nan University, Guangzhou, China (JNUKY-2021-018). All methods were performed in accordance with relevant guidelines and regulations.

In this study, we selected adolescent students aged 12–18 years from the “Family Health Index Survey in 2021”. Inclusion criteria: (1) Age 12–18 years. (2) Currently enrolled as students. (3) Having the nationality of the People’s Republic of China. (4) China’s permanent resident population (annual travel time ≤1 month). (5) Participated in the study voluntarily and filled in the informed consent form. (6) Participators could complete the network questionnaire survey by themselves or with the help of investigators. (7) Understanding the meaning of each item in the questionnaire. Exclusion criteria: (1) Inconvenient movement, confusion, mental disorders. (2) Those who are participating in other similar research projects. (3) People who are unwilling to cooperate. A total of 693 adolescent students aged 12–18 years were included in the study.

### 2.2. Measures

#### 2.2.1. The Chinese version of short-form of Family Health Scale (FHS-SF Chinese version)

The Chinese version of short-form of family health scale (FHS-SF Chinese version), by Wang et al. (2020) [42] was used in this study, which covered four dimensions (10 items): “family social and emotional health processes” (3 items: e.g., “In my family, we support each other”, etc.), “family healthy lifestyle” (2 items: e.g., “In my family, we help each other make healthy changes”, etc.), “family health resources” (3 items: e.g., “In the past 12 months, my family did not have enough money at the end of the month after bills were paid”, etc.), and “family external social support” (2 items: e.g., “In my family, we have people outside of our family we can turn to when we have problems at school or work”, etc.). (Additional information they provided on the items is available in the Appendix A). We used a 5-point scoring system from 1 to 5: “family social and emotional health processes”, “family healthy lifestyle”, and “family external social support” dimensions were scored in 1 meaning “strongly disagree”, and 5 meaning “strongly agree”. The “family health resources” dimensions were scored in reverse, with 1 meaning “strongly disagree” and 5 meaning “strongly agree”. The Cronbach’s alpha coefficient for the family health scale in this study was 0.853. The four dimensions of Cronbach’s alpha coefficients for the “family social and emotional health processes”, “family healthy lifestyle”, “family health resources”, and “family external social support” were 0.893, 0.841, 0.744, and 0.699, respectively. This showed that the internal consistency of the question items of the external social support dimension of the family in the questionnaire reached the minimum acceptable value.

#### 2.2.2. Adolescent Health Risk Behaviors Scale (AHRBS)

We used a self-designed adolescent health risk behaviors questionnaire, which was referred to the American YRBS (Youth Risk Behavior Surveillance) [5]. In this study the self-designed AHRBS had nine items, the first seven items logically measuring the frequency of occurrences leading to unintentional harmful behavior and intentional harmful behaviors in the past 30 days: swimming in unguarded water, non-compliance riding (e.g., running red lights, etc.), not wearing a seatbelt while riding, leaving home, being subjected to physical violence, non-compliance walking (e.g., using smartphones while walking, etc.), and eating unhygienic food (e.g., food that has been contaminated with bacteria and has undergone partial spoilage, etc.) (More information that they provided on the items is available in the Appendix A). We used a 5-point scoring system from 1 to 5, 1 for not applicable and 0 time, 2 for 2–3 times, 4 for 4–5 times, and 5 for ≥6 times. The last two items are the substance addiction-related entries for measuring the age of first complete exposure to addictive substances (tobacco, alcohol), the age of first smoking, and age of first alcohol consumption. We also used a 5-point scoring system from 1 to 5, 1 for “non-smoker/non-drinker”, 2 for “≤12 years old”, 3 for “13–14years old”, 4 for “15–16 years old”, and 5 for “17–18 years old”. The Cronbach’s alpha value of the AHRBS in this study was 0.735, indicating that the internal consistency of the questionnaire was favorable.

#### 2.2.3. Control Variables

We found that current stage of schooling [43,44], family economic income [45,46], and the frequency of personal computer use (iPad) [47,48] potential impacts on adolescent students’ health risk behaviors by reading the literature review. Pearson-related analysis results showed that current stage of schooling, family economic income, and frequency of personal computer use (e.g., iPad) were significantly related to the investigated variables, as shown in Table 1; thus, they were considered to be control variables in the study. The current stage of schooling was coded as primary school for 1, junior high school for 2, senior middle school for 3, and technical secondary school for 4. The frequency of personal computer use was coded as: 1 for never use, ≤1 day was 2, 2–3 days was 3, 4–5 days was 4, and 6–7 days was 5. Family economic income was coded as: 1 for ≤1500 yuan, 2 for 1501–6000 yuan, 3 for 6001–10,500 yuan, 4 for 10,501–15,000 yuan, and 5 for ≥ 15,001 yuan.

### 2.3. Data Analyses

#### Statistical Analyses

Based on the data obtained from the questionnaires, SPSS 26.0 (SPSS, Chicago, IL, USA) was used for statistical analyses, and the significance level was set at *P* < 0.05 in this study. We checked the missing values, outliers of potential variables, and hypotheses that violate normality, linearity, multicollinearity, and homoscedasticity [49]. We used the Harman single-factor test to perform factor analysis on all variables combined in the questionnaires. The results of this study showed that the unrotated principal component analysis suggested that four factors with characteristic roots greater than 1, and the variation could explain 23.542% by the first principal component, it was below the critical value (40%), even did not exceed half of the total variance explanation 58.390% [50], indicating that there was no common method bias effect among the measured variables. Secondly, we used Pearson correlations to analyze the correlations between the parameters and the predictive factors. According to the instruction of Pourhoseingholi et al. (2012) [51], we used linear regression to control the confounders. Finally, we used Mplus 8.3 to construct structural equation model diagram. We assessed the model goodness of fit (comparative fit index (CFI; good fit > 0.90), Tucker-Lewis index (TLI; good fit > 0.90), and root mean square error of approximation (RMSEA; acceptable fit < 0.08)) through the guidelines of Hu and Benteler (1999) [52]. Statistically significant horizontal structural equation models were developed at *P*-values < 0.5. The bootstrapping procedure was used to obtain estimates of total, direct, and indirect effects. For calculating bias-corrected 95% confidence intervals, 5000 bootstrapping iterations were requested [53]. We used standardized values to interpret the results.

Therefore, we applied the structural equation model and analyzed the direct and indirect impacts of the frequency of smartphone use on adolescent students‘ family health and health risk behaviors by the effect values. The frequency of smartphone use (e.g., iPad) and control variables were set as exogenous observed variables and family health supports, family health resources were set as endogenous latent variables, and the health risk behaviors in adolescent students were set as endogenous observed variables.

## 3. Results

### 3.1. Socio-Demographic Information

Socio-demographic information showed that the 693 adolescent students covered 99 primary school students, 215 middle school students, 359 high school students, and 20 technical secondary school students (they still live at home). There were 315 males and 378 females. 215 adolescent students lived in rural areas; 478 adolescent students lived in towns. 86 adolescent students had a monthly per capita household income less than 1500 yuan, 405 adolescent students had 1501 to 6000 yuan monthly per capita household income, 150 adolescent students had 6001–10,500 yuan monthly per capita household income, 26 adolescent students had 10,501–15,000 yuan monthly per capita household income, and 26 adolescent students had a monthly per capita household income that was above or equal to 15001 yuan.

In the past week, 408 adolescent students used smartphones for 6–7 days, 113 adolescent students used smartphones for 4–5 days, 92 adolescent students used smartphones for 2–3 days, 48 adolescent students used smartphones for less than 1 day, and 32 adolescent students never used smartphones. The detailed distribution of smartphone use for each demographic characteristic variable is shown in Table 2.

### 3.2. Preliminary Analyses

Descriptive statistics and Pearson analysis were performed for each of the main variables, and the results are shown in Table 1. Pearson analysis showed that the four dimensions of frequency of smartphones use and family health were related to health risk behaviors of adolescent students. The frequency of smartphones use was significantly negatively correlated to adolescents swimming in unguarded water, leaving home, and being subjected to physical violence (*P* < 0.05 and *P* < 0.01), and positively correlated with the age of first alcohol consumption (*P* < 0.01). Each dimension of family health was significantly related to the frequency of the smartphone use and teenage students’ health risk behaviors (*P* < 0.05 or *P* < 0.01). Current stage of schooling was positively related to non-compliance riding behavior and the age at which a cigarette was first smoked (*P* < 0.01). Family socio-economic status was negatively related to the behaviors of eating unhygienic food (*P* < 0.05). The frequency of personal computer use (e.g., iPad) was positively related to non-compliance walking behavior (*P* < 0.05).

### 3.3. Indices of Structural Equation Model

In this paper, we used Mplus8.3 to build the initial model. The results showed that the model indicators were X^2^/DF = 3.390; CFI *=* 0.958; TLI = 0.905; RMSEA = 0.059; SRMR = 0.048. This means that the model fit well and was statistically significant in Table 3.

### 3.4. Effect of Exogenous Latent Variables on Endogenous Variables

Adolescent students’ frequency of smartphone use was significantly and positively related to the family society and emotional health process (*β* = 0.138, CI; 0.045 0.232), family healthy lifestyle (*β* = 0.201, CI: 0.101 0.293, non-compliance walking behavior (*β* = 0.176, CI: 0.011 0.266), and the age of first alcohol consumption (*β* = 0.167, CI: 0.067 0.287), as shown in Figure 3.

In a control variable, current stage of schooling was significantly and positively associated with non-compliance walking behavior (*β* = 0.154, CI: 0.080 0.249) and the age of first alcohol consumption (*β* = 0.200, CI: 0.119 0.266). Current stage of schooling was significantly and negatively associated with family social and emotional health processes (*β* = −0.130, CI: −0.216 −0.057), family healthy lifestyle (*β* = −0.157, CI: −0.245 −0.074), and family external social support (*β* = −0.126, CI: −0.211 −0.042). There was no statistically significant correlation between family economic income on the health risk behaviors of adolescent students, but there was an association with family social and emotional health processes significantly and positively (*β* = 0.138, CI: 0.062 0.211), family healthy lifestyle (*β* = −0.099, CI: −0.016 −0.179), and family health resources (*β* = 0.310, CI: 0.229 0.392). There were no statistically significant impacts on the frequency of personal computer use (e.g., iPads) on the endogenous variables, as shown in Figure 3.

### 3.5. Effects among Endogenous Variables

The structural equation figure showed that family healthy resources were negative related to swimming in unguarded water (*β* = −0.125, CI: −0.235 −0.005), leaving home (*β* = −0.177, CI: −0.295 −0.072), and being subjected to physical violence (*β* = −0.134, CI: −0.248 −0.016), as shown in Figure 3.

### 3.6. Testing for the Mediating Effects of Family Health

After adjusting the confounders, the results showed that family health resources played a mediating role in the relationship between the adolescent students’ frequency of smartphone use, and swimming in unguarded water, leaving home, and being subjected to physical violence; the 95% confidence intervals were (CI: −0.042 −0.002), (CI: −0.049 −0.005), and (CI: −0.043 −0.002). Family healthy resources only indirectly mediated [54] the relationship among adolescent students’ frequency of smartphone use and swimming in unguarded water (the proportional values of indirect-only mediated effect is 11.2%), leaving home (the proportional values of indirect-only mediated effect is 12.4%), and being subjected to physical violence (the proportional values of indirect-only mediated effect is 11.5%), as shown in Table 4.

## 4. Discussion

Our investigation focused on adolescent students’ health risk behaviors, smartphone use, and family health. By constructing a structural equation model, we found that the frequency of smartphone use had significant positive predictive effects on adolescent students’ family health. The more frequently the adolescent students used smartphones, the healthier their self-rated families were. This finding is consistent with earlier studies, Mesch (2003) [55] studied Israeli parents and adolescents and found that parents and adolescents were able to communicate online even when they were separated. Mesch (2006) [56] conceptualized social technology as an external dynamic functioning as a break in the family boundary allowing non-familial influences to pervade, or, as a link within the family boundary drawing members closer together. More interestingly, we found that smartphone use also promoted adolescent students’ family health lifestyles, family health resources, and family external social supports. To our knowledge, this finding was not previously available.

The frequency of smartphone use was significantly and positively associated with adolescent students’ health risk behaviors, which reflected in non-compliance walking behaviors (e g., using phones while walking) and the age of first alcohol consumption. It showed that the more frequently adolescents used smartphones, the more frequently non-compliance walking behaviors (e.g., using phones while walking) occurred, and the older they were when they first consumed alcohol. Previously, many studies had associated the addition of cell phones to many dangerous behaviors [57,58]. It has demonstrated that using a smartphone frequently may lead to non-compliance walking behaviors (e g., using phones while walking). However, it can delay the adolescent students’ age of first alcohol consumption, maybe due to their media literacies. Zhang et al. [59] found that adolescent students with higher media literacy were less likely to have the intention to drink alcohol and therefore may delay the age of first alcohol consumption.

There was a negative association between family health and health risk behaviors among adolescent students, which was manifested in the family health resource dimension, as well as swimming in unguarded water, leaving home, and being subjected to physical violence. It suggested that adolescent students with more family health resources were less likely to swim in the unguarded water, leave home, or be subjected to physical violence. The previous studies suggested that the four dimensions of family health were closely associated with the behaviors of adolescents, and family social emotional processes had significant impacts on health behaviors of adolescents [60,61,62]. Healthy family lifestyles were strongly associated with reduced substance use and increased physical activity among adolescents [63]. Inadequate internal and external family resources may increase the likelihood of adolescent mental health problems [64], and mental health problems may lead to the development of adolescent students’ health risk behaviors [65]. However, none of the three dimensions of family health in this study, namely “family social emotional health processes”, “family healthy lifestyle”, and “family external social supports”, had statistically significant effects on adolescents’ health risk behaviors, which differed from previous studies. It suggested that the influence of family social and emotional health processes, family health lifestyles, and family external social support on the behavioral dimensions of adolescents could be further explored.

Smartphones can affect indirectly adolescent students’ swimming in unguarded water, leaving home, and exposure to physical violence by positively affecting family health resources. Adolescent students who used smartphones more frequently and had abundant family health resources were less likely to engage in health risk behaviors, such as swimming in unguarded water, leaving home, and being subjected to physical violence. Family health played a mediating role in the process. More specifically, family health resource dimensions of family health played indirect-only mediated roles between the frequency of smartphone use and swimming in unguarded water (the proportional values of indirect-only mediated effect is 11.2%), leaving home (the proportional values of indirect-only mediated effect is 12.4%), and being subjected to physical violence among adolescent students (the proportional values of indirect-only mediated effect is 11.5%). Combined with a theoretical perspective, the mediating role of family health cannot be underestimated. Technology-related family health may be an important moderator in distinguishing between risks and benefits [26]. Adolescent students with abundant family health resources were less likely to engage in active and passive health risk behaviors, even with frequent smartphone use. Our finding confirmed the findings of the National Survey of Children’s Health (NSCH), demonstrating that the children of families without adequate health resources were more likely to develop health risk behaviors in adulthood [66]. A good family health environment has positive impacts on the promotion of adolescent students’ healthy behaviors [67], and this perspective is also shared by national behavioral nutrition experts [68].

In addition, the study found that the more educated Chinese adolescent students were, the more likely they were to commit the non-compliance walking behaviors (e.g., using phones while walking, etc.). It may depend on the different levels of Internet penetration in different areas of education. According to the data survey conducted by the China Internet Network Information Center on the Internet penetration rates of adolescents at different education levels, the Internet penetration rates of primary school, junior middle school, high school and technical secondary college students were 92.1%, 98.1%, 98.3%, and 98.7%, respectively. As educational attainment rose, so did the popularity of the Internet. In addition, smartphones were the most used devices for Internet access, accounting for 92.2% [69]. Mourra et al. [70] found that the frequency of smartphone use of adolescents was positively correlated with using phones while walking. It suggested that the higher the level of education, the greater the likelihood of frequent smartphone use and the greater the likelihood of non-compliance walking behaviors. Current stage of schooling was positively associated with the age of first alcohol consumption, it meant that the higher the level of schooling at the current stage, the lower the risk of premature alcohol consumption, which may be related to active academic engagement. It had been observed that adolescent students who were actively involved in academics had a high degree of academic dedication [71]; they immersed themselves in their studies and spent time and energy to improve academic performance, which in turn reduced the occurrence of alcohol use [72].

### 4.1. Suggestions

Based on the findings of this current study and an extensive literature review of previous studies, we have suggested several preliminary interventions for health risk behaviors among adolescent students. We have also consulted and given feedback to stakeholders, such as: adolescent students, parents, and school leaders, to ensure that the recommendations were feasible.

We recommend that parents guide the adolescents’ smartphone use appropriately and evaluate the children’s practice of instruction and shared Internet time, rather than being “controlling”, for example monitoring efforts in the family environment. They need to recognize smartphones as an extra resource to communicate with the children in early adolescence. School is an important place where adolescents learn and socialize daily, and the school is as important as the family in intervening in the occurrence of adolescent health risk behaviors. Therefore, schools should offer various forms of relevant courses to increase the interest of young students and their parents’ participation. For example, schools can conduct on-site simulations of health risk behaviors that are likely to occur in various scenarios, invite parents of teenager students to study with them, make the courses mandatory, and hold regular health risk behavior prevention and response competitions. Educational authorities play a functional role in guiding and monitoring the effective implementation of relevant regulations and curriculum in the schools under their jurisdiction. Authorities should pay close attention to the health risk behaviors of adolescents, and develop strategies to intervene between schools and families, as well as between schools and students.

### 4.2. Limitations and Implications

The present study still has some shortcomings, and future studies need further improvement. First, this study showed significant effects of smartphone use on adolescents’ health risk behaviors. However, we considered only the frequency of smartphone use among adolescents and the types of media used by smartphones; the group of adolescents were not included. Future research could attempt to differentiate the specific content of adolescent smartphone use. Secondly, this study is a cross-sectional study, and it is difficult to state a direct causal relationship among variables. Future research could investigate the causal relationship between smartphone exposure and the prevention of adolescents’ health risk behaviors through longitudinal comparative studies in depth. Finally, the community studied in this research is only made up of adolescent students aged 12–18 years nationwide, therefore it is limited in its representation of the entire adolescent population. Future studies could cover a more diverse and complete adolescent community.

Despite limitations, the results of the current study have important practical implications. One of the important research implications of this study may be that it is the first study to explore the mediating role of family health in the relationship between smartphone use and adolescent students’ health risk behaviors by constructing a structural equation model. Family health had indirect-only mediation between frequency of smartphone use and some health risk behaviors of adolescent students. Examining family health may help build a knowledge base that can be used to map phenotypes associated with smartphone use and interventions for adolescent students’ health risk behaviors.

## 5. Conclusions

The frequency of smartphone use had positive and negative impacts on the occurrence of adolescent students’ health risk behaviors, respectively. Family health played indirect-only mediated roles between the frequency of smartphone use and some health risk behaviors of adolescent students. The current study advanced our understanding of the mechanism underlying the connection between smartphones and health risk behaviors of adolescent students. It is helpful to develop further intervention measures to prevent health risk behaviors in adolescent students by intervening in health risk behaviors that can improve the overall health of adolescent students and reduce the occurrence of health risk behaviors.

## Figures and Tables

**Figure 1 ijerph-19-13378-f001:**
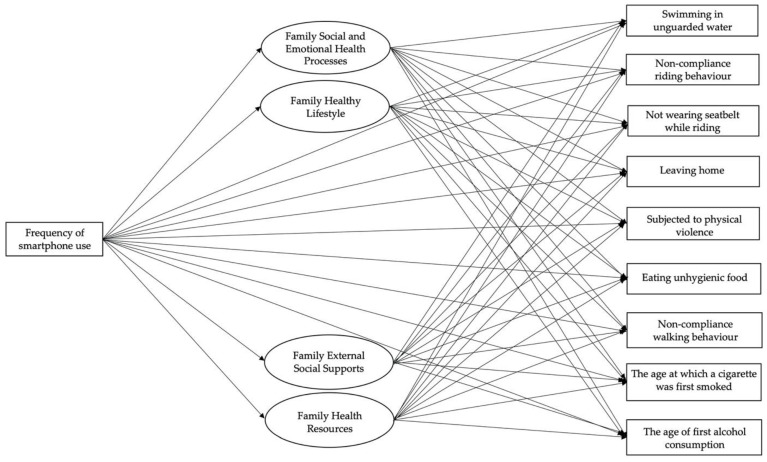
Hypothetical model.

**Figure 2 ijerph-19-13378-f002:**
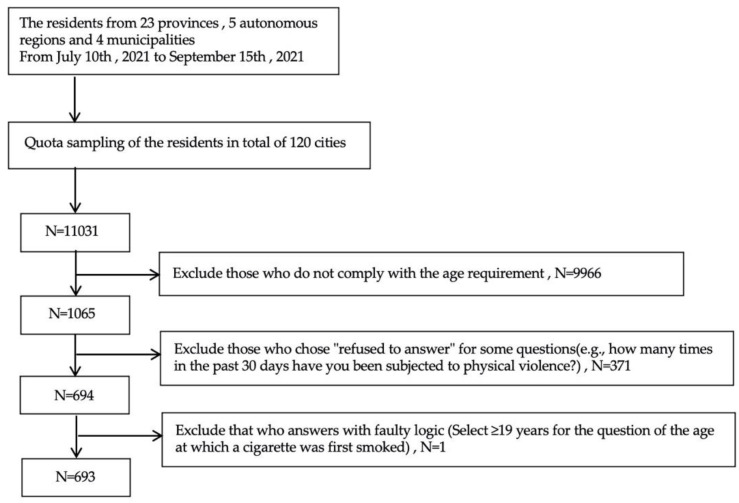
Flowchart of the data survey for this study.

**Figure 3 ijerph-19-13378-f003:**
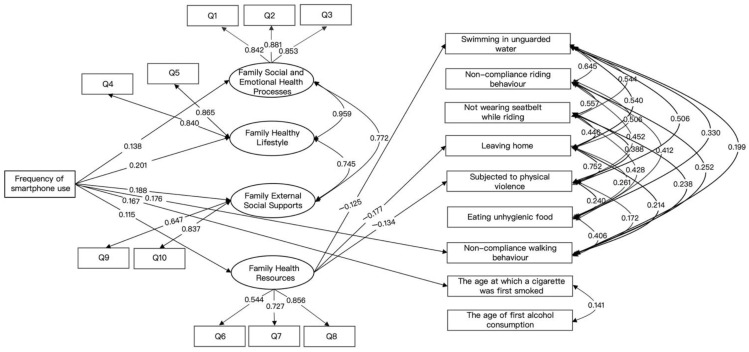
The final model. Notes: Only significant standardized estimates were presented, *P* < 0.05; current stage of schooling, family economic income, and frequency of the personal computers use (e.g., iPad) were considered as exogenous variables which were introduced into this model. They were not shown in the model for model viewing aesthetics.

**Table 1 ijerph-19-13378-t001:** Descriptive statistics, Pearson correlation coefficients of all variables.

	M ± SD	1	2	3	4	5	6	7	8	9	10	11	12	13	14	15	16	17
1. CSOS		1.000																
2. FEI		0.095 *	1.000															
3. FOPCU	4.18 ± 1.176	0.186 **	0.078 *	1.000														
4. FOSU	2.94 ± 1.405	0.180 **	0.187 **	0.365 **	1.000													
5. FSAEHP	11.90 ± 2.500	−0.076 *	0.119 **	0.102 **	0.136 **	1.000												
6. FHL	8.09 ± 1.652	−0.099 **	0.075 *	0.060	0.165 **	0.830 **	1.000											
7. FHR	11.15 ± 2.868	−0.031	0.261 **	0.017	0.085 *	0.267 **	0.271 **	1.000										
8. FESS	7.70 ± 1.664	−0.042	0.068	0.159 **	0.209 **	0.615 **	0.590 **	0.181 **	1.000									
9. SIUW	1.51 ± 0.811	0.058	−0.015	0.062	−0.077 *	−0.068	−0.072	−0.125 **	−0.049	1.000								
10. NRB	1.71 ± 0.972	0.024	−0.027	0.036	−0.030	−0.036	−0.045	−0.092 *	−0.045	0.649 **	1.000							
11. NWSWR	1.76 ± 1.031	0.032	0.009	0.053	−0.013	−0.078 *	−0.035	−0.070	−0.023	0.545 **	0.547 **	1.000						
12. LH	2.13 ± 0.557	0.022	−0.009	0.022	−0.126 **	−0.175 **	−0.161 **	−0.192 **	−0.130 **	0.556 **	0.509 **	0.458 **	1.000					
13. STPV	2.19 ± 0.645	−0.034	−0.059	−0.018	−0.121 **	−0.206 **	−0.179 **	−0.168 **	−0.105 **	0.514 **	0.448 **	0.403 **	0.764 **	1.000				
14. NWB	2.78 ± 1.747	0.210 **	0.060	0.173 **	−0.002	−0.081 *	−0.057	−0.038	−0.040	0.196 **	0.244 **	0.251 **	0.208 **	0.154 **	1.000			
15. EUF	2.23 ± 1.299	0.040	−0.080 *	−0.003	−0.002	−0.105 **	−0.070	−0.034	−0.051	0.331 **	0.404 **	0.440 **	0.271 **	0.258 **	0.399 **	1.000		
16. TAAWACWFS	1.04 ± 0.274	0.023	0.004	0.108	0.023	−0.104 **	−0.116**	−0.032	−0.101 **	0.041 **	0.092 *	0.080 *	−0.005	0.016	0.070	0.011	1.000	
17. TAOFAC	1.32 ± 0.752	0.228 **	0.030	0.046	0.164**	−0.122 **	−0.141 **	−0.035	−0.070	−0.002	0.051	0.049	0.027	0.015	0.193 **	0.088 **	0.206 **	1.000

Notes: * *P* < 0.05, ** *P* < 0.01; M = Mean; SD = Standard deviation; CSOS = Current stage of schooling; FEI = Family economic income; FOSU = Frequency of smartphone use; FOPCU = Frequency of personal computer use (e.g., iPad); FSAEHP = Family social and emotional health processes; FHL = Family healthy lifestyle; FHR= Family health resources; FESS = Family external social supports; SIUW = Swimming in unguarded water; NRB = Non-compliance riding behavior; NWSWR = Not wearing seatbelt while riding; LH = Leaving home; STPV = Subjected to physical violence; NWB = Non-compliance walking behavior; EUF = Eating unhygienic food; TAAWACWFS = The age at which a cigarette was first smoked; TAOFAC = The age of first alcohol consumption.

**Table 2 ijerph-19-13378-t002:** Summary of the demographic variables and Cross-figure of each demographic variable and the frequency of smartphone use.

			Using Time of Smartphone per Week				
Variables	Item	Total	Never Use	≤1 Day	2~3 Days	4~5 Days	6~7 Days
	Number	693	32	48	92	113	408
	Percent (%)	100%	4.6%	6.9%	13.3%	16.3%	58.9%
Gender							
Male	Number	315	12	25	54	56	168
	Percent (%)	45.5%	3.8%	7.9%	17.1%	17.8%	53.3%
Female	Number	378	20	23	38	57	240
	Percent (%)	54.5%	5.3%	6.1%	10.1%	15.1%	63.5%
Location							
Rural area	Number	215	12	21	27	44	111
	Percent (%)	31.1%	5.6%	9.8%	12.6%	20.5%	51.6%
Town	Number	478	20	27	65	69	297
	Percent (%)	69.0%	4.2%	6.0%	14.0%	14.4%	62.1%
Current stage of schooling							
Primary school	Number	99	13	12	13	22	39
	Percent (%)	14.3%	13.1%	12.1%	13.1%	22.2%	39.4%
Junior high school	Number	215	12	14	30	38	121
	Percent (%)	31.0%	5.6%	7.0%	14.0%	17.7%	56.3%
Senior middle school	Number	359	6	22	47	53	231
	Percent (%)	51.8%	1.7%	6.1%	13.1%	14.8%	64.3%
Technical secondary school	Number	20	1	0	2	0	17
	Percent (%)	2.9%	5.0%	0	10%	0	85.0%
Monthly per capita family income							
≤1500	Number	86	3	10	15	13	45
	Percent (%)	12.4%	3.5%	11.6%	17.4%	15.1%	52.3%
1501–6000	Number	405	14	29	54	70	238
	Percent (%)	58.4%	0.3%	7.2%	13.3%	17.3%	58.8%
6001–10,500	Number	150	14	7	18	21	90
	Percent (%)	21.6%	9.3%	4.7%	12%	14%	60%
10,501–15,000	Number	26	1	2	0	5	18
	Percent (%)	3.8%	3.8%	7.7%	0	19.2%	69.2%
≥15,001	Number	26	0	0	5	4	17
	Percent (%)	3.8%	0	0	19.2%	15.4%	65.4%

**Table 3 ijerph-19-13378-t003:** Assumed model-fitting indicators.

Fitting Indicators	X^2^	DF	X^2^/DF	RMSEA	SRMR	CFI	TLI
Standard			≤3 good fit; ≤5 reasonable fit	≤0.08	≤0.05 good fit; ≤0.07 reasonable fit	≥0.9	≥0.9
results	372.911	110	3.390	0.059	0.048	0.958	0.905

**Table 4 ijerph-19-13378-t004:** Path indicators and proportions of mediating effects of family health.

Path	S.C.	S.E.	T.E.	D.E.	I.E.	P.E.	95% Confidence Intervals	
							Lower Limit	Upper Limit
FOSU—FHR—SIUW	−0.014	0.009	−0.125 **	-	−0.014 *	11.2%	−0.042	−0.002
FOSU—FHR—LH	−0.020	0.011	−0.161 **	-	−0.020 *	12.4%	−0.049	−0.005
FOSU—FHR—STPV	−0.015	0.010	−0.130 *	-	−0.015 *	11.5%	−0.043	−0.002

Notes: Only an indirect path with Empirical 95% confidence interval is presented, and it doesn’t overlap with zero; Bootstrap sample size = 5000; * and ** represent statistically significant at 10% level and 5% level respectively. “-” indicates no statistically significant correlation; S.C. = Standardized coefficient; S.E. = Standard errors; T.E. = Total effects; D.E. = Direct effects; I.E. = Indirect effects; P.E. = Proportional values of indirect-only mediated effects; FOSU = Frequency of smartphone use; FHR = Family healthy resources; SIUW = Swimming in unguarded water; LH = Leaving home; STPV = Subjected to physical violence.

## Data Availability

Data are available, upon reasonable request, by emailing: bjmuwuyibo@outlook.com.

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
