# Peer review of "The Role of Family Health in Mediating the Association between Smartphone Use and Health Risk Behaviors among Chinese Adolescent Students: A National Cross-Sectional Study"

_ijerph, 2022, doi:10.3390/ijerph192013378_

Round 1
Reviewer 1 Report (New Reviewer)
This study focused on building "a structural equation model to explore the role of family health in mediating the relationship between the frequency of smartphone use and health risk behaviors", particularly with a Chinese adolescent population.
Suggestions and questions (answers could/should be used to improve the manuscript):
1. Explain figure 1. Table 1 is not cited in the text, not even explained.
2. Consider "The purpose of this study is to reduce the potential harm of smartphone use on adolescent health risk behaviors by exploring the extent to which family health variables moderate the negative effects of smartphone use."; How can a study 'reduce' that?
3. Line 321, 322, 323 is presenting figure 1, but I think it is regarding table 1.
4. What is the novelty of this study? This should be explicitly declared in the discussion section.
5. What are the suggestions presented in the discussion section based on? Authors' experience?
6. Consider "community studied in this study is only adolescent students aged 12-18 years nationwide and is not representative of the entire adolescent community". Why is it not representative? <12 and >18 would not be adolescent, right?
7. What are the implications of this study?
Author Response
Please see the attachment.

Reviewer 2 Report (New Reviewer)
Thank you for the opportunity to review this manuscript. My feedback is as follows.
Title
1. The authors should avoid using the term"impact" in the title since the study was cross-sectional in design and did not involve random allocation. The authors can use the term "association" in this regard. In addition, as I read the abstract, the authors might revise the main title as follows, "The role of family health in mediating the association between smartphone use and health risk behaviour..."
Abstract
2. The authors mentioned the "full mediated role" of family health. It would be better if the authors could mention the proportions mediated or how much the association between smartphone use and health risk behaviour can be explained by family health.
Introduction
3. For one of the hypotheses, "H2: The frequency of smartphone use will have a significant positive predictive effect 147 on family health."; does this imply that the more frequent smartphone use, the better family health of participants?
Methods
4. Since most of the participants were under 18, how did the authors obtain informed consent? Who did provide informed consent for them?
5. The authors need to mention in the main document that they provided more information on the items used to measure family health and health risk behaviours that are available in the supplementary materials.
6. In the data analysis sub-section, the authors need to explain how confounders were controlled in the analysis and whether the findings presented in the result section were adjusted for confounders.
Results
7. The authors need to calculate proportions mediated and add these in Table 5.
8. Both Tables 4 and 5 present the same thing. The authors can combine both tables into one.
9. The authors need to elaborate on whether findings from mediation analysis (SEM) presented in Tables 4 or 5 were adjusted for confounders.
Round 2
Reviewer 1 Report (New Reviewer)
The authors answered my questions, and addressed concerns. However, one point remains:
My comment: 5. What are the suggestions presented in the discussion section based on? Authors' experience?
Authors' Response Point 5: Thanks for your professional revision work on our article. The suggestions presented in the discussion section were based on three main aspects: 1) An extensive literature review of previous studies and the findings of this study;2) The project team held internal discussions based on our study findings and made several preliminary recommendations; 3) We consulted and gave feedback with stakeholders (Such as: adolescent students, parents, School Leaders) to ensure that the recommendations were feasible.
My reply: Your answer should be presented in the manuscript. Importantly, you must make it clear that the suggestions were made from the authors' vision, and also based on points 1 and 3 mentioned.
Author Response
Please see the attachment.

This manuscript is a resubmission of an earlier submission. The following is a list of the peer review reports and author responses from that submission.
Round 1
Reviewer 2 Report
The article presents a study of social interest, from the perspective of education and health. It articulates variables with an impact on the lives of adolescents, their families and the social environment.
It is scientifically sound and uses comprehensive and up-to-date bibliographical references. The figures included help to understand the concepts, results and data analysis.
The method is identified and the statistical treatment used is explained and is appropriate for the statistical analysis carried out. It supports the results and conclusions.
The researchers are aware of the limitations of the study and present them as a motto to encourage future studies.
Recommendation, accept after minor revision: In line 122 there is an allusion to Deepl Translator which the authors had better delete.
Round 2
Reviewer 1 Report
Dear Authors
I wanted you to have your English proofreading properly. but the present version was not improved yet.
Furthermore, the minimum rules that should be followed as an article are not being followed.
Please RE-RE-check your whole article again and carefully. And, please read "instructions for authors" thoroughly.
Sincerely
Author Response
Dear Reviewer,
We really appreciate all your insightful comments and suggestions, which have enabled us to grately improve our work. Based on the instructions provided in your letter, we uploaded the file of the revised manuscript ID: ijerph-1953922. Accordingly, we have uploaded a copy of the original manuscript with all the changes highlighted in colored text.
Appended to this letter is our point-by-point response to the comments and suggestions. The comments are reproduced and our responses are given directly afterward.
We are sure to have satisfactorily improved our manuscript and sincerely hope that it can be accepted for publication. Thanks again for the time and effort that you have put into reviewing our manuscript!
